# The Effect of Black Pepper (*Piperine*) Extract Supplementation on Growth Performance, Nutrient Digestibility, Fecal Microbial, Fecal Gas Emission, and Meat Quality of Finishing Pigs

**DOI:** 10.3390/ani10111965

**Published:** 2020-10-25

**Authors:** Vetriselvi Sampath, Sureshkumar Shanmugam, Jae Hong Park, In Ho Kim

**Affiliations:** Department of Animal Resource and Science, Dankook University, No. 29 Anseodong, Cheonan 330-714, Choongnam, Korea; suve2314@gmail.com (V.S.); sureshbiogenetic@gmail.com (S.S.); atom1965@hanmail.net (J.H.P.)

**Keywords:** black pepper, growth performance, nutrient digestibility, meat quality, finishing pig

## Abstract

**Simple Summary:**

Recently, Livestock industries use many herbs and spices as feed additives for various purposes. In addition, herbal extracts can be efficiently used as growth promoters, anti-stress agents, Immune boosters, and antimicrobials. However, limited studies have been conducted to evaluate the effect of black pepper extract (BPE) in swine diets. So, in this study, we use different doses of BPE (*piperine*) as supplemental additives to evaluate the growth performance of finishing pigs. The findings suggested that the graded level of BPE supplementation in pigs diet had a positive effect on growth performance, nutrient digestibility, fecal microbial, fecal gas emission, and meat quality. The results of this research will provide a new perception on the applications of BPE as a feed additive in livestock feed industry.

**Abstract:**

The study was conducted to assess the effect of black pepper extract (BPE) supplementation on the growth performance, nutrient digestibility, fecal microbial, fecal gas emission, and meat quality of finishing pigs. A total of 180 crossbred [(Landrace × Yorkshire) × Duroc] finishing pigs with average initial body weight (BW) of 53.7 ± 1.42 kg were used in 10-week trial and allotted to 6 dietary treatments (6 replications pens/treatment with 5 pigs per pen). The dietary treatments were: CON (basal diet), TRT1-CON + 0.025% BPE, TRT2-CON + 0.05% BPE, TRT3-CON + 0.1% BPE, TRT4-CON + 0.2% BPE, TRT5-CON + 0.4% BPE. Linear increase in body weight gain (BWG) (*p* = 0.038, 0.006) and average daily gain (ADG) were observed (*p* = 0.035, 0.007,and 0.006 respectively), during the overall trial in pigs fed increasing levels of BPE in supplemented diet compared to control. The dietary supplementation of BPE showed a linear increase (*p* = 0.007) in gain-to-feed ratio (G:F) at week 10. However, there were no significant results observed on average daily feed intake (ADFI) during the overall experiment. The total tract digestibility of dry matter (DM) was linearly improved (*p* = 0.053) with graded levels of BPE. In addition, BPE diet supplementation had linearly increased fecal *Lactobacillus* counts (*p* = 0.048) and decreased *Escherichia coli* counts (*p* = 0.031) in pigs at week 10. Furthermore, NH3, methyl mercaptans, and acetic acid was linearly decreased (*p* = 0.023, 0.056, 0.054) in pigs fed graded level of BPE supplementation. The inclusion of BPE in pigs’ diet had linearly increased (*p* = 0.015) backfat thickness at week 10. Thus, we concluded that BPE supplementation had positively enhanced the growth performance, nutrient digestibility, fecal microbial, fecal gas emission, and meat quality of finishing pigs.

## 1. Introduction

Antibiotics as growth promoters (AGP) are widely used in livestock feed for several years. Feeding AGP plays a beneficial role in animal production. However, Kunin [1] emphasized that routine use of AGPs in animal diets causes serious health hazard to consumers due to antimicrobial resistance. These health issues led Europe countries to prohibit the use of AGP in animal feed, since 2016 [2]. The ban on the inclusion of antibiotics as growth promoters in livestock feed urged scientists to explore a suitable alternative feed additive which enhance growth performances. In this 21st century several feed additives are used as a substitute for AGP such as probiotics [3] prebiotics [4], organic acids [2], and phytogenic [5], and these feed additives have recently gained more interest, especially in swine and poultry production [6].

Phytogenic feed additives (PFAs) are derived from plants and used in animal feed to enhance their performance. The spices *Piper nigrum* which is known as black pepper (BP) has obtained from pepper plant and belongs to the family *Piperaceae* and genus *Piper*. PFA such as spices, essential oils, herbs or plant extracts are usually blended in animals’ diets especially in swine diet [7] to improve the aroma and palatability. Hanczakowska et al. [8] noted that plant-based feed additives have potential antimicrobial properties against several pathogens. BP can be used as best alternative medicine to treat the ailments in humans including bronchitis and asthma [9]. Chopra et al. [10] stated that consumption of BP had involved in curing several health issues such as fever, asthma, cold, cough, and other general health disorders. Previously, Cheng et al. [11] reported that inclusion of herbs in swine diets had improved growth performance, nutrient digestibility, immune function, and meat quality. In 2015, Dhama et al. [5] noted that PFA has enhanced performance, feed conversion ratio, carcass meat safety and quality in animals. In addition, Windisch et al. [6] stated that herbs can improve productivity of animal. EI-Hamss et al. [12] reported that *piperine* supplementation had reduce the diarrhea problem in mice. Moreover, Yan et al. [13] observed that inclusion of BP had a significant effect on the growth performance of weaning and growing pigs. However, with a contradictory statement Cardoso et al. [14] reported that body weight and liver weight of broilers were unaffected when they are orally administrated with BP supplementation. So far, to the best of our knowledge, the effects of the BPE as feed additives have not yet been vigorously evaluated in swine diet. Thus, our study was conducted to assess the effect of dietary BPE on the growth performance, nutrient digestibility, fecal score, fecal microbial shedding, gas emission, and meat quality on finisher pigs.

## 2. Materials and Method

The protocol for this experiment was reviewed and approved by the Animal Care and Use Committee of Dankook University (DK-1-1940), Cheonan, Republic of Korea, for animal experimentation. The black pepper (BPE) extracted was provided by Synergen Company (190, Sinheung-ro, Bucheon-si, Gyeonggi-do, Korea).

### 2.1. Experimental Design, Animal Housing, and Feeding Regimen

A total of 180 crossbred [(Landrace × Yorkshire) × Duroc] finishing pigs with an average initial body weight of 53.7 ± 1.42 kg were used in 10 week trail. Pigs were randomly allotted to 6 treatments. Dietary treatments were as follows
CON (basal diet)TRT1-CON + 0.025% BPTRT2-CON + 0.05% BPTRT3-CON + 0.1% BPTRT4-CON + 0.2% BPTRT5-CON + 0.4% BP

There were 6 replicates per treatment with 5 pigs (three gilts and two barrows) per pen. All the pigs were housed in an environmentally controlled room with a slatted plastic floor and the basal diets were formulated to meet or exceed the NRC [15] recommendation (Table 1). These dietary treatments were performed during finisher pigs’ phase 1 (0–5 week) and phase 2 (6–10 week). Throughout the experimental period, each pen was equipped with a self-feeder and a nipple drinker to allow the pigs ad libitum access to feed and water.

### 2.2. Growth Performance

On day 1, at week 5, and week 10 finishing pigs body weight (BW), feed consumption and residual were weighed and recorded on a pen basis to monitor average daily gain (ADG), average daily feed intake (ADFI) and gain-to-feed (G:F) ratio.

### 2.3. Nutrient Digestibility

Chromic oxide as an indigestible marker was added in pigs’ diet to determine the apparent total tract digestibility (ATTD) of dry matter (DM), nitrogen (N), and gross energy (GE). Pigs diets were mixed with chromic oxide 7 days earlier to collect samples and fresh excreta samples were randomly collected from 2 pigs per pen at week 10 and stored at −20 °C until analyzed. Before starting the chemical analysis, the fecal samples were thawed and dried at 60 °C for 72 h. All feed and fecal samples were finely ground to pass through a 1-mm screen size and analyzed for DM and N following the procedures outlined by the AOAC [16]. Chromium was examined through UV absorption spectrophotometry (Shimadzu, UV-1201, Shimadzu, Kyoto, Japan). The GE was determined by measuring heat of combustion in the samples, using a bomb calorimeter (Parr 6100; Parr Instrument Co., Moline, IL, USA). Nitrogen content was determined by using a Kjeltec 2300 Analyzer (Foss Tecator AB, Hoeganaes, Sweden). ATTD = {1 − [(Nf × Cd)/(Nd × Cf)]}, formula was used to determine ATTD. Herewith, Nf stands for (nutrient concentration in feces), Nd stands for (nutrient concentration in diet), Cd stands for (chromium concentration in diet), and Cf stands for (chromium dioxide concentration in feces).

### 2.4. Fecal Microbial

On day 1 and at the end of the experiment, fresh fecal samples were collected by rectal massage of 2 pigs in each pen. Then the samples were pooled and placed in icebox and taken to laboratory for further analysis. One gram of excreta sample was diluted with 9 mL of 1% peptone broth (Becton, Dickinson and Co., Franklin Lakes, NJ, USA) and homogenized. Samples were then serially diluted 10-fold in 1% peptone solution and then plated onto MacConkey agar plates (Difco Laboratories, Detroit, MI, USA) and *Lactobacilli* medium III agar plates (Medium 638, DSMZ, Braunschweig, Germany) to isolate *E. coli* and *Lactobacillus*, respectively. The *Lactobacilli* medium III agar plates were incubated under anaerobic conditions at 39 °C for 48 h and the MacConkey agar plates were also incubated for 24 h at 37 °C. The colonies were counted immediately as soon as removing the plates from the incubator.

### 2.5. Fecal Noxious Gas Emission

The stock feces (300 g) were placed in plastic boxes with a small hole in a middle and sealed with a plaster. The samples were fermented for one day at room temperature (25 ℃), and 100 mL of sampled was taken from the headspace (approximately 2.0 cm) above the fecal sample for the air circulation. Later the box was re-sealed to measure the fecal noxious content. Collected fecal samples were manually shaken around 30 s to measure the crust formation on the surface and homogenized. Concentrations of NH_3_, H_2_S, methyl mercaptan, and acetic acid were measured within the scopes of 5.0 to 100.0 ppm (No. 3La, detector tube; Gastec Corp. Kanagawa, Japan) and 2.0 to 20.0 ppm (4LK, detector tube; Gastec Corp).

### 2.6. Back-Fat Thickness and LMP

On day 1, week 5, and week 10 the back-fat thickness (BFT) and lean meat percentage (LMP) of all pigs were measured using a real-time ultrasound instrument (Piglog 105; SFK Technology, Herlev, Denmark). The mean value of LMP and BFT was recorded for statistical analysis.

### 2.7. Meat Quality

To evaluate physicochemical properties, all pigs were slaughtered at a local commercial slaughterhouse. Following exsanguinations and evisceration, pigs’ muscle was collected from the dressed carcass 30 min postmortem and chilled at −20 °C before subsequent analysis. Lean meat samples were thawed at room temperature before evaluation. The meat color, marbling, and firmness scores were evaluated according to National Pork Producers Council [17]. Each sample surface was measured at three locations to detect lightness, redness, and yellow values (Model CR-410 Chromameter, Konica Minolta Sensing Inc., Osaka, Japan). About 10 g of fresh meat was minced and mixed with 90 mL distilled water and blended in tissue homogenizer. Digital pH meter was used to record the pH suspension. The pH meter probe was measured using two buffers (pH 4.0 and 7.0) and each measurement was repeated 3 times. The water-holding capacity (WHC) was calculated by the techniques of Kauffman et al. [18]. Of the sample, 0.3 g was pressed at 26 °C for 3 min with 125 mm diameter filter paper. Areas of the compressed sample and the expressed humidity were defined and determined by using a digitalized area-line sensor (MT-10S, M.T. Precision Co. Ltd., Tokyo, Japan). The ratio of water:meat area was then calculated, giving a measure of WHC (a smaller ratio indicates increased WHC). The Longissimus muscle area was measured by locating the LM surface in the 10th rib, which was conducted using the digitization area-line sensor. Drip loss was measured with plastic bag method described by Honikel [19] using approximately 2 g a meat sample. Cooking loss was determined by the methods of Sullivan et al. [20].

### 2.8. Statistical Analysis

All data were examined statistically in a completely randomized design using mixed procedures of SAS (SAS Inst. Inc., Cary, NC, USA) with pen as the experimental unit. Orthogonal comparisons were conducted using polynomial regression to determine Linear and quadratic effects on the graded level of 0%, 0.025%, 0.05%, 0.1%, 0.2%, and 0.4% of BPE extract in pigs diet. Variability in the data was expressed as the standard errors mean. Differences among treatment means were determined using Tukey’s range test for overall *p*-value. The *p* < 0.05 was considered as significant and *p* < 0.10 was considered as trend.

## 3. Results and Discussion

In recent decades, natural spices have become the appealing food supplement in animal feeding. Usually, feed is used for ensuring the dietary energy of animals. BPE has been widely known for its anti-microbial properties and the extract from its fruits and leaves poses strong anti-bacterial activity against plant pathogenic [21]. In 2011 [13], Yan et al. report that herbal extract mixture which includes BPE had a beneficial effect on the growth performance of growing pigs. Moreover, Czech et al. [22] showed that mixed herbal extract (garlic, licorice roots and tiller, thyme herb, and caraway fruits) had enhanced the growth performance of pigs. In this study, Table 2 illustrate the inclusion of dietary BPE supplementation on pigs BW, ADG, ADFI, and G:F. Initially no significant differences were observed on pigs’ body weight. However, pigs fed increasing level of BPE in diet had linearly increased (*p* = 0.038, 0.006) the body weight at week 5 and 10 compared to control. These findings are consistent with that of Ghazalah et al. [23] and Mansoub [24] who stated that broilers fed BPE supplementation had improved body weight. The ADG was linearly increased (*p* = 0.035, 0.007, 0.006) in pigs at week 5, 10 and overall experiment due to graded level of BPE supplementation compared then control. Moreover, at the end of the trial, Pigs fed BPE supplementation had significantly increased (*p* = 0.007) G:F compared than CON diet. This result was consistent with Al-Kassie et al. [25] who reported that broilers supplemented with hot pepper had enhanced the G:F. In addition, Cardoso et al. [14] reported that the broilers fed 60 mg/kg BPE supplementation showed a better G:F conversion ratio. However, there were no significant differences observed on ADFI during the overall experiment. In addition, pigs supplemented with a BPE additive count not influence the ADG, ADFI, and G:F in overall *p*-value and this outcome agrees with Al-Kassie et al. [25] who observed no significant difference on broiler fed diet supplemented with BPE.

In 2007 [26], Srinivasan stated that *piperine* had beneficial effect on therapeutic drugs and phytochemicals through several mechanisms. BPE can promote pancreatic digestive enzymes such as lipase, amylase and proteases, and plays vital roles in the digestion process [27]. At week 10, the increased level of BPE supplementation on finishing pigs’ diet had linearly increased (*p* = 0.053) the nutrient digestibility of DM compared than control diet (Table 3). Similarly, Yan et al. [13] reported that the supplementation of herbal extract had increased the growth performance and the ATTD of DM in growing pigs. However, Nousiainen and Setala [28] noted a contradictory statement that pigs nutrient digestibility was not affected by the inclusion of herbal plant mixture (cinnamon) at week 10. Moreover, our results failed to show significant difference on digestibility of N and E indicating that BPE supplementation does not affect nutrient digestibility of finishing pigs. Previously, Ao et al. [29] stated that a mature digestive system and enhanced immunity helps in fighting against intestinal disorder of an older pigs and that may result in the lack of nutrient digestibility. This line was correlated with our study that well-developed digestive system and enhanced immunity of finishing pigs may be considered as one of the reasons for the lack of significant difference in nutrient digestibility of N and E. Furthermore, feed formulation, BPE dosage levels, and pigs age may also be the reason for this lack of significant results.

The fecal microbes play a major role in detoxifying harmful substances, preventing colonization of pathogens, recycling the nitrogen, and synthesis microorganisms of vitamins [30]. The increased level of BPE supplementation in finishing pigs’ diet had linearly increased *lactobacillus* counts (*p* = 0.048) and decreased *E. coli* (*p* = 0.031) counts at week 10 compared to control (Table 4). This finding agreed with the results of Yan et al. [31] who found decreased *E. coli* counts in pig fed BPE supplementation. In addition, EI-Hamss et al. [12] reported that *piperine* supplementation had reduce the diarrhea problem in mice. Thus, we believe that the anti-microbial effects of BPE may be the reason for the reduction of *E. coli.* counts in pigs.

The fecal noxious gas content such as NH_3_ and H_2_S concentration, are the major components of pig manure which cause the air pollution [32]. The elevated level of NH_3_ and H_2_S may cause health hazard to animals as well as humans [33]. Fecal gas emission content is generally associated with the nutrient digestibility and increased digestion which may lower substrates of microbial fermentation in large intestine and resulted in reduced gas emission content [34]. However, at the end of this trail, NH_3,_ Methyl mercaptans and acetic acid emission was linearly decreased (*p* = 0.023, 0.056, 0.054) in pigs fed BPE supplementation compared to CON. This finding was agreed with Wenk [35] who suggested that the inclusion of herbal extract could reduce the fecal noxious gas content in growing pigs by controlling the microflora of the intestinal tract. However, there were no significant results found on H_2_S and Co_2_ emission at overall experiment (Table 5). Hence this experiment reveals that decreased NH_3_ gas emission in finishing pigs, maybe due to the increase of digestibility and intestinal microflora balance.

Pigs’ back fat thickness (BFT) is an important predictor of carcass lean content and meat quality [36]. Previously, Grzes et al. [37] stated that the impact of BFT has reflection on lean meat percentage and helps to evaluate the meat quality. Moreover, Boyd et al. [38] found BFT as an important parameter to determine the amount of energy in pig within a short span of time. Roongsitthichai and Tummaruk [39] insist that the sow reproductive performance was determined by the backfat thickness. The BFT and Lean meat percentage of finishing pigs fed BPE supplementation is presented in Table 6. During week 10, the backfat thickness was linearly increased (*p* = 0.015) but there was no significant difference found on LMP on pigs fed BPE supplementation. This result was agreed with B. Matysiak et al. [40] who observed that plant extract mixture had a significant effect on backfat thickness at weaning. Moreover, Kwon et al. [41] suggest that plant mixture has positive effects on meat quality of growing–finishing pigs. However, Ilsley et al. [42] observed a contradictory result, that dietary plant extracts (capsicum, carvacrol, cinnamaldehyde) had no significant difference on backfat thickness in lactating sows.

The effect of BPE supplementation diet on finishing pigs’ meat quality is shown in Table 7. Apart from backfat thickness the water-holding capacity (WHC) and intramuscular fat (IMF) are also considered to be the most important traits of meat quality [43]. Besides, drip loss is another important visual sign to assess quality of meat. Ngapo et al. [44] said that the weight loss due to drip loss usually ranges from 2% to 10% when the meat is cut into slices. Besides, the study of Balasubramanian et al. [45] reveals that drip loss reduction had improved meat quality and this saying was linked with the current study that pigs fed BPE supplementation diet had tendency to decrease the drip loss (*p* = 0.057) on d3 compared to CON.

Aaslyng et al. [46] indicated that juiciness of the meat was determined by a combination of the water content, IMF content and the saliva production during chewing the pork. Previously, Rincker et al. [47] indicate that there was strong relationship between IMF content and the sensory traits of “juiciness” and “tenderness” in pork. However, there was no interactive effect found on water holding capacity, cooking loss, meat color (lightness, redness, and yellowness values), and sensory evaluation. These inconsistent findings about meat quality may be due to the increased level of BPE supplementation or differences in experimental conditions. We believe that further research will assist to find the exact reason for the lack of significant results on water holding capacity, cooking loss, and meat color.

## 4. Conclusions

Plant based additives are not only acting as palatants, but also have a positive impact on other physiological functions which helps to maintain good health and improved performance. The limited number of studies available on BPE have shown some favorable results on the growth and production. In this regard, we conclude that the inclusion of black pepper supplementation could enhance the growth performance, nutrient digestibility, fecal microbial, fecal gas emission, and meat quality of finishing pigs. Moreover, this finding will provide a new perception on the applications of BPE as a feed additive in livestock feed industry.

## Figures and Tables

**Table 1 animals-10-01965-t001:** Composition of finishing pig diets (as fed-basis).

Items	Phase 1 (0–5 Week)	Phase 2 (6–10 Week)
Ingredients (%)		
Corn	37.98	36.15
Wheat	24	29
Rice bran	2	2
Parm kernell meal	3	3
Soybean meal	3	3
Dehulled Soybean meal	11.34	8.12
Rape seed meal	4	4
Sesame meal	2	2
Brown Rice	5	5
Animal fat	3.26	2.89
Molasses	2	2
Limestone	1.08	1.1
Monocalcium phosphate	0.1	0.09
Salt	0.3	0.3
Methionine 98%		0.01
Threonine 98%	0.01	0.05
Lysine 25%	0.49	0.79
Choline Chloride 50%	0.09	0.1
Vitamin/Mineral mixture ^1,2^	0.35	0.4
Total	100.00	100.00
Chemical composition		
Digestible Energy (kcal/kg)	3540	3510
Metabolic Energy (kcal/kg)	3260	3250
C.Protein (%)	16.00	15.00
C.Fat (%)	5.90	5.50
C.Ash (%)	4.20	4.10
C.Fiber (%)	3.90	3.90
Total Lysine (%)	0.88	0.86
Calcium (%)	0.65	0.65
Phosphorus (%)	0.39	0.39

^1^ Provided per kg diet: Fe, 115 mg as ferrous sulfate; Cu, 70 mg as copper sulfate; Mn, 20 mg as manganese oxide; Zn, 60 mg as zinc oxide; I, 0.5 mg as potassium iodide; and Se, 0.3 mg as sodium selenite. ^2^ Provided per kilograms of diet: vitamin A, 13,000 IU; vitamin D_3_, 1700 IU; vitamin E, 60 IU; vitamin K_3_, 5 mg; vitamin B_1_, 4.2 mg; vitamin B_2_, 19 mg; vitamin B_6_, 6.7 mg; vitamin B_12_, 0.05 mg; biotin, 0.34 mg; folic acid, 2.1 mg; niacin, 55 mg; D-calcium pantothenate, 45 mg.

**Table 2 animals-10-01965-t002:** The effect of Black pepper supplementation on the growth performance of finishing pigs ^1^.

Items	CON	TRT1	TRT2	TRT3	TRT4	TRT5	SEM ^2^	*p*-Value ^3^		Overall *p*-Value
								Linear	Quadratic	
Body weight, kg										
Initial	53.73	53.73	53.72	53.71	53.71	53.71	0.01	0.202	0.757	1.000
Week 5	81.58	81.52	81.71	81.91	82.29	82.53	0.40	0.038	0.540	0.001
Week 10	112.99	113.17	113.44	113.85	114.96	115.86	0.80	0.006	0.356	0.001
Week 5										
ADG, g	796	794	800	806	817	823	11	0.035	0.561	1.880
ADFI, g	2260	2245	2265	2291	2302	2326	27	0.521	0.975	0.744
G/F	0.349	0.348	0.350	0.352	0.356	0.359	0.058	0.230	0.679	0.935
Week 10										
ADG, g	897	904	907	913	933	952	15	0.007	0.341	0.102
ADFI, g	2823	2822	2827	2837	2839	2848	29	0.453	0.879	0.798
G/F	0.318	0.320	0.321	0.322	0.329	0.334	0.042	0.007	0.297	0.201
Overall										
ADG, g	847	849	853	859	875	888	11	0.006	0.355	0.088
ADFI, g	2553	2551	2557	2564	2571	2573	17	0.264	0.877	0.303
G/F	0.332	0.333	0.334	0.335	0.340	0.345	0.03	0.866	0.320	0.300

^1^ Abbreviation: CON (basal diet), TRT1-CON + 0.025% BPE, TRT2-CON + 0.05% BPE, TRT3-CON + 0.1% BPE, TRT4-CON + 0.2% BPE, TRT5-CON + 0.4% BPE. ^2^ Standard error of means. ^3^ Means in the same row with different superscript differ significantly (*p* < 0.05).

**Table 3 animals-10-01965-t003:** The effect of black pepper supplementation on the nutrient digestibility of finishing pigs ^1^.

Items, %	CON	TRT1	TRT2	TRT3	TRT4	TRT5	SEM ^2^	*p*-Value ^3^		Overall *p*-Value
								Linear	Quadratic	
Week 10										
Dry matter	71.18	71.51	71.64	71.92	72.78	73.98	1.03	0.053	0.451	0.711
Nitrogen	70.99	69.05	69.73	70.61	72.02	72.73	1.03	0.067	0.117	0.385
Energy	70.78	71.25	71.53	72.31	71.71	72.92	0.94	0.122	0.984	0.434

^1^ Abbreviation: CON (basal diet), TRT1-CON + 0.025% BPE, TRT2-CON + 0.05% BPE, TRT3-CON + 0.1% BPE, TRT4-CON + 0.2% BPE, TRT5-CON + 0.4% BPE. ^2^ Standard error of means. ^3^ Means in the same row with different superscript differ significantly (*p* < 0.05).

**Table 4 animals-10-01965-t004:** The effect of black pepper supplementation on the fecal microbial of finishing pigs ^1^.

Items, log10cfu/g	CON	TRT1	TRT2	TRT3	TRT4	TRT5	SEM ^2^	*p*-Value ^3^		Overall *p*-Value
								Linear	Quadratic	
Week 10										
*Lactobacillus*	7.51	7.53	7.60	7.63	7.63	7.64	0.05	0.048	0.483	0.261
*E. coli*	6.24	6.20	6.15	6.14	6.11	6.08	0.05	0.031	0.823	0.052

^1^ Abbreviation: CON (basal diet), TRT1-CON + 0.025% BPE, TRT2-CON + 0.05% BPE, TRT3-CON + 0.1% BPE, TRT4-CON + 0.2% BPE, TRT5-CON + 0.4% BPE. ^2^ Standard error of means. ^3^ Means in the same row with different superscript differ significantly (*p* < 0.05).

**Table 5 animals-10-01965-t005:** The effect of black pepper supplementation on the fecal gas emission of finishing pigs ^1^.

Items, ppm	CON	TRT1	TRT2	TRT3	TRT4	TRT5	SEM ^2^	*p*-Value ^3^		Overall *p*-Value
								Linear	Quadratic	
Week 10										
NH_3_	1.68	1.90	1.70	1.25	1.33	1.23	0.21	0.023	0.787	0.143
H_2_S	5.93	6.03	5.90	5.85	5.80	5.75	0.22	0.404	0.828	0.820
Methyl mercaptans	4.38	4.20	3.85	3.93	3.75	3.68	0.21	0.056	0.540	0.206
Acetic acid	9.13	9.03	8.75	8.70	8.68	7.95	0.40	0.054	0.570	0.089
CO_2_	11,000	11,000	10,000	9750	9250	9000	1417	0.347	0.801	0.752

^1^ Abbreviation: CON (basal diet), TRT1-CON + 0.025% BPE, TRT2-CON + 0.05% BPE, TRT3-CON + 0.1% BPE, TRT4-CON + 0.2% BPE, TRT5-CON + 0.4% BPE. ^2^ Standard error of means. ^3^ Means in the same row with different superscript differ significantly (*p* < 0.05).

**Table 6 animals-10-01965-t006:** The effect of black pepper supplementation on backfat thickness and lean meat percentage (LMP) of finishing pigs ^1^.

Items	CON	TRT1	TRT2	TRT3	TRT4	TRT5	SEM ^2^	*p*-Value ^3^		Overall *p*-Value
								Linear	Quadratic	
Initial										
BFT, mm	9.0	9.2	9.3	9.5	9.6	9.7	0.3	0.109	0.797	0.966
LMP, %	65.0	65.0	65.1	65.5	65.5	65.2	0.2	0.241	0.378	0.889
Week5										
BFT, mm	13.5	13.7	13.9	14.0	14.0	14.4	0.3	0.090	0.942	0.974
LMP, %	58.1	58.0	57.9	57.9	58.1	58.2	0.2	0.642	0.188	0.954
Week10										
BFT, mm	18.0 ^b^	18.3 ^ab^	18.3 ^ab^	18.6 ^ab^	18.5 ^ab^	18.9 ^a^	0.3	0.015	0.909	0.564
LMP, %	54.2	54.5	54.5	54.4	54.7	54.6	0.2	0.062	0.456	0.627

^1^ Abbreviation: CON (basal diet), TRT1-CON + 0.025% BPE, TRT2-CON + 0.05% BPE, TRT3-CON + 0.1% BPE, TRT4-CON + 0.2% BPE, TRT5-CON + 0.4% BPE. BFT—backfat thickness, LMP—lean meat percentage ^2^ Standard error of means. ^3^ Means in the same row with different superscript differ significantly (*p* < 0.05).

**Table 7 animals-10-01965-t007:** The effect of black pepper supplementation on meat quality of finishing pigs ^1^.

Items	CON	TRT1	TRT2	TRT3	TRT4	TRT5	SEM ^2^	*p*-Value ^3^		Overall *p*-Value
								Linear	Quadratic	
pH	5.42	5.39	5.33	5.40	5.37	5.30	0.04	0.832	0.507	0.354
WHC, %	35.03	37.84	38.52	36.42	35.44	36.85	1.99	0.330	0.438	0.789
Cooking loss, %	31.74	30.99	29.95	31.49	30.62	29.26	1.76	0.756	0.570	0.421
Drip loss, %										
d1	8.18	8.75	8.45	8.44	8.40	8.21	0.96	0.075	0.182	0.138
d3	14.02	13.73	13.17	13.52	13.21	13.10	0.62	0.057	0.884	0.486
d5	18.57	18.60	17.69	18.47	18.03	17.03	0.47	0.372	0.174	0.279
d7	23.48	22.21	22.22	23.44	22.63	22.32	0.40	0.139	0.668	0.582
Meat color										
L*	47.09	47.93	47.32	47.46	47.06	47.11	0.67	0.074	0.218	0.219
a*	14.20	14.42	14.32	14.49	14.29	14.42	0.43	0.285	0.947	0.876
b*	4.31	4.35	4.36	4.42	4.44	4.46	0.22	0.969	0.239	0.291
Sensory evaluation										
Color	3.24	3.23	3.22	3.25	3.25	3.19	0.18	1.000	0.248	0.643
Firmness	3.25	3.22	3.19	3.28	3.22	3.21	0.12	0.824	0.448	0.471
Marbling	3.13	3.16	3.22	3.16	3.22	3.22	0.12	0.590	0.959	0.892

^1^ Abbreviation: CON (basal diet), TRT1-CON + 0.025% BPE, TRT2-CON + 0.05% BPE, TRT3-CON + 0.1% BPE, TRT4-CON + 0.2% BPE, TRT5-CON + 0.4% BPE. WHC-Water holding capacity. ^2^ Standard error of means. ^3^ Means in the same row with different superscript differ significantly (*p* < 0.05).

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
