# Peer review of "The Effect of Black Pepper (Piperine) Extract Supplementation on Growth Performance, Nutrient Digestibility, Fecal Microbial, Fecal Gas Emission, and Meat Quality of Finishing Pigs"

_animals, 2020, doi:10.3390/ani10111965_

Round 1

Reviewer 1 Report

Dear authors

I can not review the manuscript by missing line number in current vision, animals-930259-peer-review-v1. The manuscript must be rewritten carefully.

Author Response

Response to the Reviewer(s)' Comments: 

We heartily thank for reviewer’s valuable comments. We had carried out all the suggested correction as per the reviewer’s comments throughout the manuscript and the edited context were highlighted in the red-colored text.

Thank you for considering the revised version of our manuscript for possible publication in animals. We appreciate reviewers and editors for their comments and suggestions, and we believe that these comments are highly constructive and very useful to restructure the manuscript. We have thoughtfully revised the manuscript and highlighted them in text. The response to reviewers’ concerns is given point by point mentioned below. We hope that all these changes fulfill the requirements to make the manuscript acceptable for publication in animals.

Reviewer 1:

I cannot review the manuscript by missing line number in current vision, animals-930259-peer-review-v1. The manuscript must be rewritten carefully.

Response: As per the reviewer’s comment, we have included the missing line number in the revised manuscript. We thank the reviewer for this suggestion, which helped to increase the readability of our manuscript.

Reviewer 2 Report

The paper is interesting and in my opinion needs only minor changes.

See the joined file to see the details. 

Author Response

Response: We would like to thank reviewers for making valuable comments, as per the reviewer’s comments we resolved all the corrections/comments/suggestions throughout the manuscript, the text of the entire manuscript was revised by a language professional as per the necessary action.

Reviewer 3 Report

Summary: The aim of the paper is clear and there is a potential interest for these type of feed additives, due to the importantness of finding alternatives to antibiotic growth promotors. The objective of the paper is clear and well substantiated as there is need for more research in the area.

Broad comments: The introduction and material & methods are linguisticly well written, although it is suggested to get it proof red for language check.

However, the part on material & methods needs to be better described in terms of clarification. The main weakness of the paper is that it lacks information on the procedure of experimental design, e.g. the part on allocation to treatments and blocks. Further, the paper needs clarification in the statistical model used. E.g which models were used? Fixed and random effects? And were possible interactions included in the model?

According to the material & method, the different dependent variables were based on either pen-basis or individual measures. This must be taken into account in the statistical model and corrected for. This type of data might also be better analysed with p-differences, instead of linear regression, as it could be of interest to see where differens appear, or at which inclusion level an effect could be obtained.

This inadequacy might have resulted in not reliable results, and must therefore be taken into concideration. Therefore, I will not comment on the result and discussion part of the paper. Consequently, I'm not able to judge whereas the conclusion is supported by the results.

Author Response

Summary: The aim of the paper is clear and there is a potential interest for these type of feed additives, due to the importantness of finding alternatives to antibiotic growth promotors. The objective of the paper is clear and well substantiated as there is need for more research in the area.

Response:  Authors would like to thank to the reviewers for the positive effort made towards our manuscript.

Broad comments: The introduction and material & methods are linguisticly well written, although it is suggested to get it proof red for language check.

Response: We would like to thank reviewer’s for suggesting valuable comments, as per reviewer’s comments our manuscript was revised by a language professional as per the necessary action.

However, the part on material & methods needs to be better described in terms of clarification. The main weakness of the paper is that it lacks information on the procedure of experimental design, e.g. the part on allocation to treatments and blocks. Further, the paper needs clarification in the statistical model used. E.g which models were used? Fixed and random effects? And were possible interactions included in the model?

Response: We thank the reviewer for this suggestion, which helped to increase the readability of our manuscript. As per reviewer suggestion, the Materials & methods section was rephrased. In this study pigs were allocated to one of six dietary treatments basal diet supplemented to corn- wheat-based diets with 0%, 0.025%, 0.05%, 0.1%, 0.2%, and 0.4% black pepper extract. Each treatment consisted of six replications with five pigs (three gilts and two barrows) per pen in a randomly complete block design based on gender and BW. In addition, all data were examined statistically in a completely randomized design using mixed procedures of SAS (SAS Inst. Inc., Cary, NC) with pen as the experimental unit. Orthogonal comparisons were conducted using polynomial regression to determine Linear and quadratic model and Variability in the data was expressed as the standard errors mean. Differences among treatment means were determined using Tukey’s range test for overall p-value was used in this study.

According to the material & method, the different dependent variables were based on either pen-basis or individual measures. This must be taken into account in the statistical model and corrected for. This type of data might also be better analysed with p-differences, instead of linear regression, as it could be of interest to see where differences appear, or at which inclusion level an effect could be obtained.

Response: Thank you for the reviewer’s valuable suggestion, in this case all response criteria, each pen served as the experimental unit not in individual measures. As per reviewer suggestion, we had included overall p-differences in the table section and the necessary modification were done throughout the manuscript.  

This inadequacy might have resulted in not reliable results, and must therefore be taken into concideration. Therefore, I will not comment on the result and discussion part of the paper. Consequently, I'm not able to judge whereas the conclusion is supported by the results.

Response: Thank you for your valuable suggestions/comments. As per reviewer suggestion the statistical difference were include in the tables and result parts were rephrased, and the necessary modifications were done throughout the manuscript.

Reviewer 4 Report

Topic of article is very actual. Authors did experiment with finishing pigs feeding. They used one control group and five experimental groups. Authors gained a lot of results (as is wrote in article title). Majority of determined values was without significant differences between groups, but trends was confirmed.

My comments:

Simple summary – Add information, if according results of this experiment have addition of black pepperoni to swine diets positive or negative effect on analysed parameters.

Abstract – It is not clear if was examined the effect of black pepper or the effect of black pepper extract???

Chapter 2.1 – specify what form of black pepper was used (extract, powder …)? Or, haw was mixed and homogenized in swine diets of each experimental group. If was used the extract, haw did you do this extract?

Chapter 2.1, Table 1. – Why is “wheat bran” between components of diet? No one of diet contain wheat bran.

Chapter 2.1, Table 1. – What kind of animal fat was used?

Chapter 2.1, Table 1. – What mean abbreviation “MCP”?

Chapter 2.1, Table 1. – How was determined the concentration of DE and NE in the diets?

Without these information the repeating of experiment is impossible.

Tables in chapter Results and Discussion – may by is better, when footnotes contain this information (between groups no significant difference was found) if there is no significant difference between groups.

Author Response

Topic of article is very actual. Authors did experiment with finishing pigs feeding. They used one control group and five experimental groups. Authors gained a lot of results (as is wrote in article title). Majority of determined values was without significant differences between groups, but trends was confirmed.

My comments:

Simple summary – Add information, if according results of this experiment have addition of black pepperoni to swine diets positive or negative effect on analysed parameters.

Response: Thank you for reviewer’s valuable suggestion, as per the suggestion we modified. The findings suggested that the graded level of black pepper extract supplementation in pigs diet has a positive effect on growth performance, nutrient digestibility, fecal microbial, fecal gas emission, and meat quality.

Abstract – It is not clear if was examined the effect of black pepper or the effect of black pepper extract???

Response: Thank you for reviewer’s valuable comment, we have examined this study with black pepper extract, and the necessary modification were done throughout the manuscript.

Chapter 2.1 – specify what form of black pepper was used (extract, powder …)? Or, haw was mixed and homogenized in swine diets of each experimental group. If was used the extract, haw did you do this extract?

Response: Thank you for your comment. In this study we have used black pepper extract   and it was commercially provided by the Synergen Company (190, Sinheung-ro, Bucheon-si, Gyeonggi-do, Republic of Korea). Basal diet and black pepper extract were mixed with a help of feed mixer machine in swine diet of each experimental group. We have mention the protocol for black pepper extract below.

Black pepper   Grinding   Extract with 95% ethanol (3 times)     Filtration   Concentration    Crystallization    Recrystallization     Vacuum drying  Pulverizing     Final product

Chapter 2.1, Table 1. – Why is “wheat bran” between components of diet? No one of diet contain wheat bran.

Response: It’s our pleasure to respond your valuable comments. In this study diet composition, we didn’t include wheat bran there was typographical errors. In the revised manuscript we removed wheat bran diet contain.

Chapter 2.1, Table 1. – What kind of animal fat was used?

Response: Thanks for reviewer’s valuable comments, Tallow was used as an animal fat in the study.

 Chapter 2.1, Table 1. – What mean abbreviation “MCP”?

Response: Thanks for reviewer’s valuable comments, as per your comments we have include full abbreviation monocalcium phosphate (MCP) in Table1.

Chapter 2.1, Table 1. – How was determined the concentration of DE and NE in the diets?

Without this information the repeating of experiment is impossible.

Response: We thank you for your valuable comments pinpoint by the reviewer. Hence in this study we didn’t determined the concentration of DE and ME in the composition of finishing pig diets. In this study the amount of DM and ME (kcal/kg) was used in the diet composition.

Tables in chapter Results and Discussion – may by is better, when footnotes contain this information (between groups no significant difference was found) if there is no significant difference between groups.

Response: Thank you for your valuable suggestions/comments. As per reviewer suggestion the table chapter and result and discussion parts were rephrased, and the necessary modifications were done throughout the manuscript.

Authors would like to thankful to the editors for the valuable effort made towards our manuscript. As per the editor’s and reviewer’s suggestion essential modifications can be done in the manuscript. Thank you for your kind support for publishing our manuscript in your esteemed journal.

Round 2

Reviewer 3 Report

General comments

The quality of the paper has generally been improved. The authors considered most of the reviewer comments and recommendations. On the basis of the revisions made to the paper and the relevance of the topic, I recommend that the manuscript can be published after further minor revisions and improvements of the manuscript.

Specific comments:

There is need to revise the inconsistency in the use of e.g commas, epual signs, brackets throughout the whole manuscript. Please change this.

As previously mentioned, the authours state that The P<0.05 was considered as significant and P<0.10 was considered as trend. According to this it is not acceptable to regard P=0.057 (line 239) as significant linear decrease. Please change.

It is not clearly demonstrated any changes of the information in the tables (not included in the revised version). Please go through the tables according to previous comments from reviewers. 

Line 192: Please check the format of the reference included.

Line 235: Please check the format of the reference included.

Line 241: Please check the format of the reference included.

Yours Sincerely,

Author Response

RESPONSE TO THE REVIEWER(S)

Thanks for the reviewer’s valuable comments, as the reviewer said I resolved all the corrections by using colored text throughout the manuscript.

The quality of the paper has generally been improved. The authors considered most of the reviewer's comments and recommendations. On the basis of the revisions made to the paper and the relevance of the topic, I recommend that the manuscript can be published after further minor revisions and improvements of the manuscript.

Response:  Authors would like to thank the reviewers for the positive effort made towards our manuscript. As per the reviewer’s suggestion, we had revised the manuscript thoroughly. 

Specific comments:

There is need to revise the inconsistency in the use of e.g commas, epual signs, brackets throughout the whole manuscript. Please change this.

Response: We have performed the suggested change. We thank the reviewer for this suggestion, which helped to increase the readability of our manuscript

As previously mentioned, the authors state that The P<0.05 was considered as significant, and P<0.10 was considered a trend. According to this, it is not acceptable to regard P=0.057 (line 239) as a significant linear decrease. Please change.

Response: We modified as per the reviewer’s comment. Thank you

It is not clearly demonstrated any changes in the information in the tables (not included in the revised version). Please go through the tables according to previous comments from reviewers. 

Response: We would like to thank you for the reviewer’s valuable comments, as per the reviewer’s comments we resolved all the corrections/comments/suggestions throughout the tables.

Line 192: Please check the format of the reference included.

Line 235: Please check the format of the reference included.

Line 241: Please check the format of the reference included.

Response: Thank you for your suggestion, in this regard we made the changes according to the journal format guideline and model papers from the journal of Animals.

Citations;

“Balasubramanian B, Park JH, Shanmugam S, Kim IH. 2020. Influences of enzyme blend supplementation on growth performance, nutrient digestibility, fecal microbiota, and meat-quality in grower-finisher pigs. Animals. 2020 Feb 27;10(3):386. DOI: 10.3390/ani10030386”

The authors would like to thank the editors for the valuable effort made towards our manuscript. As per the editor/reviewer’s suggestion essential modifications can be done to the manuscript.

With regards

In Ho Kim